# Endothelial Senescence and Its Impact on Angiogenesis in Alzheimer’s Disease

**DOI:** 10.3390/ijms241411344

**Published:** 2023-07-12

**Authors:** Irina Georgieva, Jana Tchekalarova, Dimitar Iliev, Rumiana Tzoneva

**Affiliations:** 1Institute of Biophysics and Biomedical Engineering, Bulgarian Academy of Sciences, Acad. George Bonchev, Str. Bl. 21, 1113 Sofia, Bulgaria; igeorgieva@biomed.bas.bg; 2Institute of Neurobiology, Bulgarian Academy of Sciences, Acad. George Bonchev, Str. Bl. 23, 1113 Sofia, Bulgaria; jt.chekalarova@inb.bas.bg; 3Institute of Molecular Biology, Bulgarian Academy of Sciences, Acad. George Bonchev, Str. Bl. 21, 1113 Sofia, Bulgaria; diliev@bio21.bas.bg

**Keywords:** angiogenesis, cellular senescence, aging, extracellular vesicles, oxidative stress, Alzheimer’s disease

## Abstract

Endothelial cells are constantly exposed to environmental stress factors that, above a certain threshold, trigger cellular senescence and apoptosis. The altered vascular function affects new vessel formation and endothelial fitness, contributing to the progression of age-related diseases. This narrative review highlights the complex interplay between senescence, oxidative stress, extracellular vesicles, and the extracellular matrix and emphasizes the crucial role of angiogenesis in aging and Alzheimer’s disease. The interaction between the vascular and nervous systems is essential for the development of a healthy brain, especially since neurons are exceptionally dependent on nutrients carried by the blood. Therefore, anomalies in the delicate balance between pro- and antiangiogenic factors and the consequences of disrupted angiogenesis, such as misalignment, vascular leakage and disturbed blood flow, are responsible for neurodegeneration. The implications of altered non-productive angiogenesis in Alzheimer’s disease due to dysregulated Delta-Notch and VEGF signaling are further explored. Additionally, potential therapeutic strategies such as exercise and caloric restriction to modulate angiogenesis and vascular aging and to mitigate the associated debilitating symptoms are discussed. Moreover, both the roles of extracellular vesicles in stress-induced senescence and as an early detection marker for Alzheimer’s disease are considered. The intricate relationship between endothelial senescence and angiogenesis provides valuable insights into the mechanisms underlying angiogenesis-related disorders and opens avenues for future research and therapeutic interventions.

## 1. Introduction

Angiogenesis is a complex biological process that involves the formation of new blood vessels from preexisting ones and should not be confused with vasculogenesis, in which blood vessels emerge de novo from endothelial progenitor cells [1]. It plays a crucial role during embryonic development and later in tissue growth and repair, wound healing, and reproduction. Still, it must be carefully regulated to avoid excessive or insufficient vascularization. New vessels emerge from sprouting endothelial cells (EC), the leading players, toward an angiogenic stimulus (sprouting angiogenesis) or by insertion into existing vessels and division into new ones (splitting angiogenesis) [2]. It is led by a tip cell that elongates and explores the environment while transmitting signals to the stalk cells that follow behind to proliferate and form tubular networks. The entire process is highly complex and difficult to imitate in vitro, highlighting the need for development of reliable models to study it [3,4,5]. Angiogenesis is governed by a strict balance between pro- and antiangiogenic factors, which, if broken, leads to uncontrolled cell proliferation (cancer, atherosclerosis, rheumatoid arthritis) or mitotic inhibition (aging and neurodegenerative diseases) [6,7,8]. Excessive angiogenesis can promote inflammation and tissue damage, while insufficient angiogenesis can lead to ischemia and cell death. Most studies are focused on the involvement of angiogenesis in cancer and cardiovascular diseases (CVD). Fewer examine its contribution to neurodegeneration, although it is correlated with the impairment of angiogenesis [9], endothelial senescence [7] and the occurrence of cerebrovascular angiopathy (a process in which small blood vessels burst and cause hemorrhages) [10,11]. The altered blood circulation in the elderly contributes to the lengthy process of wound healing and inadequate recovery of ischemic tissues due to the lack of response from aged ECs. Typically, ECs’ physiological functions are suppressed in time because of accumulated stress and induction of cellular senescence and apoptosis [12], leading to alterations in the regulation of angiogenesis and insufficient or excessive vascularization [6]. Age-related vasculature dysfunction has been implicated in the pathogenesis of various neurodegenerative diseases, including Alzheimer’s disease [13], Parkinson’s disease [14], and Huntington’s disease [15]. It may contribute to their progression by modulating the delivery of nutrients and oxygen and clearing of waste products from the brain.

Alzheimer’s disease (AD) is a debilitating condition characterized by progressive cognitive decline and behavioral changes that severely impact the daily lives of suffering individuals. Similarly, to other neurodegenerative diseases, aging is an essential factor contributing to its onset. There is overwhelming research aiming to find the causes, better ways for detection, treatment and, if possible, ways to avoid it altogether (reviewed elsewhere [16,17,18,19,20]). Factors involved in angiogenesis have roles in the birth of new neurons (neurogenesis), the prevention or mitigation of neuronal injury (neuroprotection), and the pathogenesis of stroke, AD and motor neuron disease [21]. Indeed, axon and blood vessel growth and migration are braided together via chemo-repulsive and attractive signals in which the vascular endothelial growth factor (VEGF) and the Delta-Notch signaling have a direct effect on both nervous and vascular systems [22], confirming that angiogenesis is closely related to neurodegeneration. AD patients exhibit changes in the number, diameter and density of blood vessels, which lead to decreased brain perfusion and BBB disruption. Here, we explore the current debate on the effect of the monomeric, oligomeric and plaque forms of amyloid-β on the efficacy of cerebral angiogenesis and blood flow.

This narrative review further explores the intricate relationships among senescence, oxidative stress, extracellular vesicles, and the extracellular matrix, highlighting their significance in the processes of angiogenesis, aging, and neurodegeneration. Additionally, it delves into potential therapeutic approaches aimed at modulation of angiogenesis and amelioration of disease progression.

## 2. The Dual Nature of Cellular Senescence

Cellular senescence is a fundamental process associated with tissue homeostasis during development, first described by Hayflick and Moorhead [23]. The authors observed a terminal pause in cell division of normal human fibroblasts after several cycles of passaging. They concluded that cultured cells cease to proliferate upon a finite number of doublings and, therefore, could be used as a model for aging. Today, this is referred to as the Hayflick limit. The processes of senescence and aging are intertwined in the sense that aging progresses with time and associates with increased numbers of senescent cells. Therefore, cellular senescence is also accepted as a hallmark of aging and a risk factor for age-related neurodegenerative diseases. However, senescence occurs during the full lifespan of an individual and is not restricted to later life stages. The resulting inability to divide is a consequence of irreversible cell cycle arrest, caused by the accumulation of various stress factors such as DNA damage, inflammation, telomere shortening, chromatin perturbations, and oncogene induction [12,24,25,26]. Senescence is believed to have evolved as a protective mechanism against cancer, but it also contributes to age-related physiological decline [27]. Additionally, loss of senescence during embryonic development allows the progression of unhealthy cells in embryos [28]. In contrast, while protecting against the propagation of mutated DNA, senescence harms long-living organisms, as it inhibits tissue renewal and function. These observations gave rise to the idea that there is a “right time to senesce”, arguing that the end goal of the fight against aging is not to completely eliminate senescent cells (SCs) but to learn how to tame them [29].

### 2.1. Hallmarks of Aging

In 2022, several new hallmarks of aging were introduced, stressing the complexity of the aging process [25]. They include compromised autophagy, impeded RNA processing, and changes in the microbiome and in the mechanical properties of both cells and extracellular matrix [25]. However, until recently the main focus was on the hallmarks of aging involving information loss (reviewed in [30]), telomere shortening [31] and endogenous reactive oxygen species (ROS) induced DNA damage [32] (Figure 1). The first is based on the fact that even though an organism shares the same genetic DNA among all cells, the epigenetic differences between them determine the cellular type. With age, epigenetic changes procured in response to DNA damage and p53 activation make information harder to read and trigger loss of cellular identity and function. This process is similar to reprogramming somatic cells to induced pluripotent stem cells (iPSC), achieved by the four Yamanaka factors, Oct4, Sox2, Klf4 and c-Myc [33]. A recent study supporting this theory utilized inducible epigenetic modifications to instigate premature aging in transgenic mice. The subsequent introduction of the Yamanaka factors reversed the “old” phenotype, hypothetically improving the animals’ quality of life [34]. Similar treatments were able to restore vision in aged mice [33], and short exposure to the same factors (for 13 days) rejuvenated fibroblasts and allowed them to maintain their original cell identity and improved their collagen secretion [35].

The other two hypotheses do not necessarily exclude epigenetic modifications. Instead, they focus on senescence as the cell’s response to life-long accumulation of stress-induced DNA damage by telomere shortening or oxidative stress (OS). The former is the result of the end-replication problem that causes dysfunctional telomeres and triggers the DNA-damage response (DDR) through ataxia-telangiectasia mutated protein kinase (ATM), checkpoint kinase 2 (CHK2), p53-binding protein 1 (53BP1) and γ-H2AX histone [36]. T cells can overcome this issue by elongating their telomeres, using telomeres from antigen-presenting cells (APCs), rather than increased telomerase activity. The intercellular transfer of telomeres via extracellular vesicles (EVs) rescues T cells from senescence and maintains their long-term immunological memory [37]. However, telomere extension can be overwhelmed by other senile factors, as telomerase activation cannot prevent senescence caused by OS-induced DNA damage in human fibroblasts but protects them against apoptosis and necrosis [38]. Furthermore, murine cells have significantly longer telomeres than human cells, but under standard culturing conditions, they senesce substantially faster due to high O_2_ levels and severe OS [39]. The induction of senescence in otherwise normal cells through exogenous factors such as chemicals [40], EVs generated from premature senile cells [41], septic shock [42] and OS is also known as stress-induced premature senescence [43].

In addition to its role in DDR, the protein kinase ATM functions as a sensor of redox homeostasis. It is oxidized and activated by hypoxia [44], resulting in ATM-mediated phosphorylation and stabilization of hypoxia-inducible factor (HIF)-1α [45], but it can also cause premature endothelial senescence and dysfunction [46]. In one study, OS stimulated ATM through the Akt/p53/p21 pathway, causing senescence in ECs, which was not the case for ATM-KO mice or upon treatment of HUVECs with ATM inhibitors [47]. Moreover, genetic or pharmacological ATM inhibition reduced cellular senescence and SASP expression [48]. ATM, Akt and the mammalian target of rapamycin (mTORC1) activation, as part of DDR, stimulates mitochondrial biogenesis and ROS-mediated DNA damage and senescence. Both of these processes are ameliorated by ATM or mTORC1 suppression [49].

DNA repair can be affected by numerous factors, including changes between anaerobic and aerobic carbohydrate metabolism that alter NAD^+^ levels. The molecule is used as a cofactor by multiple enzymes such as sirtuins (SIRTs)—NAD-dependent protein deacetylases involved in epigenetic modifications. SIRTs are known as “longevity genes” because their overexpression resulted in life extension in yeast, and they are depleted by insufficient NAD^+^ [50]. SIRT deficiency has been correlated with stress-induced premature endothelial [51] and hepatocyte [52] senescence. In a recent review by Charles Bennet [53], the author shares his disbelief in the correlation between lifespan, NAD^+^ and SIRT. He argues that the positive outcome of NMN and NAD^+^ supplementation is not due to the activation of SIRT but rather due to the antioxidant effect of these cofactors. Moreover, NAD^+^ is necessary for poly-(ADP-ribose) polymerase 1 (PARP1) activity in DDR [36] and, therefore, reduces the extent of DNA damage. Although PARP1 participates in one of the major DNA repair mechanisms, it is also involved in parthanatos (programmed cell death, independent from caspases, unlike apoptosis) [54]. Notably, OS-induced DNA damage and cell death can be avoided with PARP1 inhibitors or PARP1-KO [54], which also reduces PARP1-associated senescence-associated secretory phenotype (SASP) [55]. Meanwhile, SIRT1 has been associated with improved endothelial function [56] and increased microvascular density [57], whereas its knockout results in decreased angiogenesis [58]. A clearer picture of the interplay between NAD^+^- (SIRT1, PARP1) and redox sensors (ATM), and their contribution to endothelial senescence, could be immensely useful.

### 2.2. The Dose Makes the Poison

The cell’s choice between senescence or death depends on the level of accumulated stress and the subsequent activation of p53 [59]. Intermediate levels of p53 signal for senescence, and its hyper- or hypoactivation causes cell death or proliferation, respectively. To choose a path forward, the cell first undergoes a cell cycle arrest. If the experienced changes persist above a certain threshold, the cell proceeds with irreversible senescence, a process called geroconversion [60]. In the case of chronic stress, senescence can be triggered through either p53/p21 or retinoblastoma (Rb)/p16 pathways [26,59]. Therefore, if a cell is positive for either of these proteins and for senescence-associated β-galactosidase (SA-β-Gal), it can be considered senescent. Accumulation of SCs with chronological age varies depending on the cell and tissue types. The same applies to the expression and the activity of the factors that control the senescence signaling pathways.

SCs exhibit a hypersecretory phenotype known as SASP, which is used to alert the immune system to reduce local inflammation by eliminating them and, potentially, to direct tissue renewal. Removing SCs in this manner is essential because most of them are resistant to apoptosis. If SASP secretion is maintained for a short period, the consequential clearance of unhealthy cells can be very beneficial. On the other hand, prolonged SASP further increases intercellular stress and has the opposite effect. The SASP factors could stimulate nearby premalignant cells’ growth and angiogenic activity and, paradoxically, promote excessive angiogenesis and subsequent progression of cancer or neurodegenerative diseases [26,61]. Unfortunately, as the immune system weakens with age, its ability to clear SCs is reduced, and SASP evolves with a change from anti- to pro-inflammatory cytokine secretion [62]. Thus, the notion that temporally regulated mechanisms orchestrate the functions of SCs is probably the most coherent senescence concept so far. While all the beneficial roles of senescence share a transient profile, the deleterious functions of SCs are associated with their lingering persistence, namely chronic exposure to their SASP [29].

## 3. Endothelial Senescence

Aging and prolonged exposure to environmental factors, such as toxins, ROS, shear stress, and extracellular matrix (ECM) perturbations, induce senescence in ECs (Figure 1). Interestingly, unlike most SCs, senescent ECs (sen-ECs) remain susceptible to apoptosis [63], a mechanism most likely evolved to rearrange the microvasculature and counteract proliferation. Senescence in ECs is usually triggered by telomere shortening [26], which can be avoided by the exogenous introduction of telomerase [6]. Ionizing radiation can also geroconvert human microvascular cells in a time- and dose-dependent manner, predominantly by uncoupling Complex II of the mitochondrial respiratory chain [64], demonstrating ECs’ susceptibility to OS. In any case, the balance between senescence and angiogenesis becomes dysregulated during aging and neurodegenerative diseases, but the underlying mechanisms remain elusive. The negative consequences of vascular aging are apparent in older people in whom the regeneration of blood flow after ischemia or wounding is a slow and tedious process [65]. The accumulated stress over time reduces the proliferative capacity of ECs and modifies their interaction with the already altered ECM [66]. Furthermore, aging reduces the general expression of vascular endothelial growth factor (VEGF) [6] and promotes angiogenic incompetence in ECs, making them unable to respond to VEGF [7]. Some of the suggested reasons for the VEGF insensitivity are the age-related loss of VEGF receptor 2 (VEGFR2) [67], androgen resistance [68] and reduction in nitric oxide (NO) [6]. Furthermore, the SASP can directly inhibit angiogenesis by secreting factors that block endothelial cell proliferation and migration. At the same time, SCs can induce angiogenesis by secreting pro-inflammatory cytokines that promote neovascularization.

### 3.1. ECM Disruption Accelerates Vascular Aging

Aged vasculature is described with increased microvascular perfusion, susceptibility toward pro-inflammation and atherosclerosis, disrupted ECM interaction and altered secretory, barrier and transport functions [8,9,25,69]. The ECM comprises the natural scaffolding and framework on which ECs reside. The latter shape the vessel’s lumen, align to its length and curvature by attaching to the basal membrane and control the permeability, contractility and passage through the vessel [70]. The ECM consists primarily of collagen, elastin and fibrinogen, synthesized by ECs and fibroblasts and subjected to constant rearrangement by resident cells. It also mediates chemical cues that can alter the cell’s response and vice versa, creating the tissue microenvironment and enabling ECs to proliferate, migrate and stimulate vascular smooth muscle cells to form capillary networks and constrict/dilate fully formed vessels. One of the hallmarks of endothelial senescence and blood vessel aging is the stiffening of the ECM through glycation, aggregation and crosslinking [71]. Therefore, diseases occur not only when the cells are damaged but also when the ECM becomes impaired. For instance, elastin is renewed quite slowly; thus, changes in its structure tend to persist for longer periods of time. Meanwhile, collagen secretion increases, causing the stiffening of ECM in an NAD^+^-dependent manner [69]. Angiogenesis also relies heavily on the deposition and degradation of the ECM by metalloproteinases (MMPs), whose activity increases with age, further reducing the elasticity of the connective tissue and stimulating higher traction forces even in non-senescent cells [71]. This leads to endothelial dysfunction characterized by excess angiogenesis, leaky vasculature and low shear stress that cannot induce protective signaling pathways—a faulty process described as non-productive angiogenesis. Hence, by treating the age-related stiffening of the ECM, we could tackle endothelial dysfunction. One such example is the treatment of myocardial tissue with an optimized intravascular infusible ECM, which is able to fill gaps between the ECs, reduce vascular leakiness and improve vascular fitness [72].

### 3.2. Navigating the Currents: Shear Stress and Its Impact on Endothelial Cells’ Function

Naturally, endothelial cells are constantly exposed to shear stress in vivo from the movement of a non-Newtonian fluid, i.e., the blood. The wall shear stress (WSS) is described as the traction forces generated on the endothelial wall by a flow and depends on its velocity [73,74,75]. This type of stress activates the endothelial NO synthase (eNOS), aids in cellular alignment and protects against endothelial dysfunction [66,76]. In a comprehensive review, Yi-Shuan J. Li et al. summarized the effect of WSS on ECs and concluded that high shear stress inhibits apoptosis through PI3K/Akt-mediated activation of eNOS and increases migration [77]. It should be noted that values of high and low WSS can vary between studies and cell lines. In some cases, excessive proliferation is stimulated by disturbed (oscillating) flow that provokes local monolayer permeability (high turnover-leaky hypothesis [78]). This brings the question of whether the overall strength of the WSS or the local occurrence of such is the triggering force for these effects (Table 1). Moreover, could the increased proliferation aim to induce new vessel formation to dissipate the high pressure, especially in aortic ECs? In contrast, active cell division could be a substantial issue for brain blood vessels, as microvascular ECs must avoid uncontrolled proliferation and non-productive angiogenesis to maintain the blood–brain barrier (BBB). In agreement, human microvascular ECs (HMVECs) do not elongate in response to increased curvature and/or shear stress, presumably to minimize the length of tight junctions (per unit length of the capillary) and reduce the paracellular transport into the brain [79]. Human umbilical vein ECs (HUVECs), on the other hand, can migrate both with and against the flow, demonstrating the interplay between function and response to environmental signals [80].

The flow also helps to form the lumen of blood vessels and organizes the ECs during sprouting angiogenesis. Recent research describes the counteracting forces between the actin cytoskeleton of ECs and the hemodynamic forces of the flow, which are necessary to establish a fully functioning vessel [82]. It is unclear how sen-ECs respond to the flow regarding lumen formation. Since sen-ECs exhibit significant morphological changes and stronger focal adhesion compared to cell–cell contacts [83], it would be interesting to study whether senescent cells can withstand these forces.

### 3.3. Linking NO Signaling with Endothelial Senescence

NO plays various roles beyond vasorelaxation, including influencing the maturation of endothelial progenitor cells, mitochondrial function, cell division, and inhibiting platelet aggregation and pro-inflammatory cytokine-induced signaling pathways. In short, NO protects against the factors contributing to endothelial senescence. It is generated from L-arginine by eNOS, which requires tetrahydrobiopterin (BH4) as a cofactor (Figure 2). The expression of eNOS is induced by shear stress [84], but its activity is inhibited by ROS or NG-nitro-L-arginine methyl ester (L-NAME). Furthermore, lack of BH4 or L-Arg can cause eNOS uncoupling and production of superoxide anion (O_2_^•−^) instead of NO—a major cause of endothelial senescence. A reaction between the superoxide anion and NO produces ONOO^−^ (peroxynitrite (PN)), further reducing NO’s bioavailability and promoting eNOS uncoupling and vascular dysfunction through OS [85]. PN also causes lipid peroxidation, protein oxidation and nitration, and LDL oxidation through Apolipoprotein E (ApoE). Inhibition of eNOS decreases the activity of human telomerase (hTERT) in HUVECs, making them susceptible to telomere-induced senescence [86]. Moreover, eNOS-KO mice experience premature cardiac aging and aortic stiffness, which is explained by increased calcium-dependent focal adhesion [87]. Although, PN-dependent Ca^2+^ influx in ECs leads to vascular dysfunction [88], when it is generated by shear stress, it activates eNOS through calcium-calmodulin complexes [89]. Undisturbed laminar flow also upregulates eNOS transcription in an ERK1/2- and NF-kB-dependent manner, contributing anti-inflammatory properties to NF-kB activation [90]. In addition, a novel eNOS modulator—MAGI1 (MAGUK with inverted domain structure-1), associated with VE-cadherin in cell-cell contacts—can support NO production under shear stress via PKA/AMPK-mediated mechanism [91]. Considering the vasoprotective properties of NO, eNOS activity under physiological flow can counteract endothelial dysfunction (Figure 2).

## 4. Unveiling the Interplay between Hypoxia and Oxidative Stress-Induced Endothelial Senescence

The main reason for O_2_’s negative manifestation is that it is responsible for the generation of reactive oxygen species (ROS), which cause DNA damage and induce senescence. It is still unclear whether there are different mechanisms of senescence activation, depending on the source of ROS and/or the place of accumulation [92]. For example, in CVD, mitochondrial dysfunction often triggers age-associated perturbations in the production of NO and VEGF [27,66], which can be mitigated by reduced mitochondrial oxidative phosphorylation in mammals [93]. On the other hand, mitochondrial ROS in the model organism *Caenorhabditis elegans* increases its longevity [94]. In addition, reduced mitochondrial mass and alterations in the electron transport chain (ETC) due to a decline in cytochrome C oxidase and Complex IV [95] and upregulated NADPH oxidases (NOX) increase OS and shorten telomeres [96]. The role of mitochondria in senescence was also confirmed by global transcriptomic analysis, where the expression of 38% of senescence-associated genes was reversed in mitochondrial-depleted fibroblasts [49]. A direct link between ROS, telomere shortening and senescence was evidenced by assessing the number of SA-β-Gal^+^ ECs after exposure to H_2_O_2_ or glutathione (GSH) peroxidase inhibitors (it should be noted that other senescence markers were not used) [96]. Since OS is a prominent contributor to endothelial senescence, it is natural to assume that low levels of O_2_ could prevent this process. Interestingly, low ROS delay DNA replication and cell cycle progression via a CDK2-dependent mechanism [97]. Therefore, lower ROS levels and prolonged cell division could potentially prevent replicative EC senescence due to excessive telomere shortening and reduced DNA damage. The following section further explores the interplay between hypoxia and OS-induced endothelial senescence.

### 4.1. HIF-1α in Angiogenesis

The excess O_2_ under typical in vitro experiments (20% pO_2_) generates a significant amount of ROS, making common culturing conditions hyperoxidative [92]. In contrast, human, bovine and murine fibroblasts grown under 5% pO_2_ increase their lifespan by 20%, 80% and up to 500%, respectively, due to significantly less OS [98,99,100]. Based on the available data, it is reasonable to consider that lower oxygen levels may reduce OS. This is not entirely true, because prolonged lack of O_2_ can seriously affect cellular metabolism and function, leading to tissue damage and organ failure if not treated promptly. This state is known as hypoxia and occurs when the body or a specific tissue or organ is deprived of adequate oxygen. However, short-term hypoxia can act as a hormetic stress (a short jump out of the individual’s comfort zone and subsequent quick recovery to homeostasis) and increase cell resilience (for a detailed review, see [101].)

Hypoxia orchestrates angiogenesis through the main pro-angiogenic factor—VEGF. Its expression is regulated by the three isoforms of HIF—1, 2 and 3α. They are under the control of prolyl-hydroxylases (PHDs), which target HIFs for proteasomal degradation but are sensitive to oxygen deprivation and are destroyed under hypoxic conditions. HIF-1α is ubiquitously expressed and responds to acute respiratory changes, whereas HIF-2α is responsible for chronic hypoxia and is localized in ECs and glial cells. Besides VEGF, other typical targets of HIF-1α and 2α are glucose transporter 1 (GLUT1) and lactate dehydrogenase A (LDHA). In addition, HIF-1α regulates the expression of erythropoietin (*Epo*) and *Mmp-9*, and HIF-2 that of *Oct4* [102]. Besides acting as a transcription factor for VEGF, HIF-1α recruits endothelial progenitor cells from bone marrow and supports their differentiation into ECs, increases the expression of VEGF receptors (VEGFR1/2), stimulates the production and secretion of MMPs and recruits supporting cells to create mature and stable blood vessels [103]. Under hypoxic conditions, EVs carrying MMPs can also stimulate ECs to proliferate, migrate, and form capillary-like structures [104,105].

### 4.2. Hypoxia Is Essential for the Day–Night Cycle

Although, HIF-1α is associated with poor prognosis in cancer and CVD, it works synergistically with one of the core regulators of the circadian rhythm—basic helix-loop-helix ARNT like 1 (BMAL1) [106]. In fact, hypoxia is so tightly related to the circadian clock that the incidence of heart attacks increases on the Monday following the daylight saving time transition [107]. Increase in HIF-1α during the day triggers the expression of the pro-angiogenic genes—*Vegf*, *Epo* and *Glut1*, but its persistence corresponds with the activation of pro-apoptotic genes—*Bnip3* and *Noxa1* in cells [106]. The circadian rhythm is also regulated by the hormone melatonin. It is synthesized predominantly by the pineal gland at night and is suppressed by bright light. Besides its role in the awake–sleep cycle, it is also one of the strongest known natural antioxidants. Melatonin supplementation can improve sleep and reduce jet lag, which is also observed with mild hypoxia [108]. Our recent studies demonstrated its ability to reduce OS and improve cognitive functions in a rat AD model [109]. Notably, short-term fluctuations in O_2_ activate autophagy, degrade damaged mitochondria and reduce mitochondrial ROS. Hence, there is an evolutionary pressure to adapt to moderately low O_2_ levels. In contrast, inhibition of autophagy during prolonged exposure to hypoxia increases the levels of ROS due to the uncoupling of complex III and generation of semiquinone (QH•). This results in O_2_^•−^ formation and its conversion to H_2_O_2_ by superoxide dismutase (SOD) [110]. The generated H_2_O_2_ inhibits PDHs and indirectly stabilizes HIF-1α, causing chronic hypoxia [103] (Figure 3). Furthermore, inhibition of the ETC by hypoxia can lead to mitochondrial dysfunction and increased EV secretion, causing inflammation in many cell types [111].

During hypoxia, HIF-1α induces the expression of glycolytic enzymes, stimulates glycolysis and inhibits pyruvate dehydrogenase kinase 1 (PDK1), suppressing the pyruvate dehydrogenase complex (PDH) and the TCA cycle. As a result, mitochondrial respiration is reduced along with mitochondrial ROS and senescence [112]. However, disruption of the circadian clock impairs anaerobic glycolysis [113], causing acidification of the cells [114]. Lower pH redirects perinuclear lysosomes to the cytoplasm’s periphery, separating mTORC1 from its upstream activators, inhibiting its activity [114]. mTORC1 is an essential sensor for nutrients and feeding times, central to establishing the circadian rhythm. Hence, environmental factors such as the ATP/AMP ratio, NAD^+^ bioavailability and the overall redox state of the cell could prepare the organism for the light cycle as well as its metabolic response. Similarly, hypoxia can modulate the cell cycle arrest triggered by p16/Rb and provoke apoptosis resistance in SCs by elevating Bcl-2, Bcl-xL and p21 levels [60]. Even though p21 overexpression is connected with senescence, moderate levels promote cell survival, as p21-KO mice accumulate significant DNA damage and undergo apoptosis [115]. Hypoxia inhibits mTOR, which suppresses the conversion from p21-mediated cell cycle arrest to irreversible senescence [116], increases NO levels [117], inhibits NF-kB and decreases SASP [118], independent of p53 and HIF-1α. Inhibition of autophagy by mTOR can lead to insulin resistance, further increasing the concentration of glucose in blood plasma, subsequent OS and endothelial senescence. This demonstrates another possible mechanism through which mTOR inhibition can suppress aging [119] (Figure 3). Nevertheless, lower mitochondrial ROS failed to reduce senescence in hyperoxic conditions unless p53 and Rb were inhibited [120].

## 5. Exploring the Role of Extracellular Vesicles in Angiogenesis and Senescence

Legends about the infamous Hungarian Countess Elizabeth Bathory tell the story of her supposed anti-aging process of bathing in the blood of young girls. A similar idea governs the myths for vampires, which might not necessarily stay young, but become immortal by feeding on human blood. Surprisingly, there seems to be some truth in these myths, as recent studies showed that blood exchange from young to old mice rejuvenates them, but the opposite transfusion leads to senescence in the young [121,122,123]. The latter highlights the role of SASP in aging, which assists the immune response and, in the context of angiogenesis, influences new vessel formation. In addition to soluble factors, such as chemokines, inflammatory cytokines and growth factors, extracellular vesicles (EVs) are key components of SASP (reviewed in [124]). EVs are a very heterogeneous group of membranous structures, roughly categorized into three main groups based on size and origin: apoptotic bodies (ABs), microvesicles (MVs) that range from 50 to 5000 nm and are formed by outward budding and fission of the plasma membrane, and exosomes (30–100 nm) that are produced by the fusion of multivesicular endosomes with the plasma membrane, releasing intraluminal vesicles into the extracellular space. The EVs play important roles in intercellular communication, and their release is a strictly regulated process [125]. They are involved in both physiological and pathological processes and play a role in intercellular communication through the transfer of proteins, lipids, and nucleic acids [126,127]. EVs are implicated in cancer etiology due to their ability to promote cancer cell migration, transformation of non-malignant cells and pro-angiogenic activity [128]. While healthy cells release EVs as part of normal cellular homeostasis, senescent cells secrete EVs that have a significant role in angiogenesis and neurodegenerative disease progression. The presence of pro-angiogenic molecules like HIF-1α, VEGF, MMPs, and microRNAs in EVs [129] may lead to homeostasis disruption and non-productive angiogenesis. The role of EVs as key functional components of SASP is further highlighted by the observation that secretion of EVs is much higher in different types of senescent cells, including ECs, as compared to young ones [130,131]. A possible explanation for this is the observed upregulation of neutral sphingomyelinase and dysfunction of lysosomal activity in senescent cells [132]. One study even suggests that hypoxia prevents senescence by decreasing the SASP, rather than reducing the number of senescent cells [118].

The important role of EVs from ECs, as well as other blood cell types, in angiogenesis is summarized here [125]. More specifically, EVs from ECs are rich in β1 integrins and metalloproteinases (MMP-2 and MMP-9), which allow them to penetrate the ECM, to remodel it and to form tubular capillary-like structures. Stimulation with VEGF and FGF-2 facilitates the association of the active and proenzyme forms of the MMPs with EC-derived vesicles [133]. EVs can also transport urokinase plasminogen activator/uPA receptor (uPA/uPAR), which are both pro-angiogenic. It was shown that uPAR modulates VEGF-induced EC migration by balancing the proteolysis of the ECM and the cell motility through integrin-associated focal adhesion (Figure 4). Revu Ann Alexander and colleagues demonstrated that VEGF causes endocytosis of αVβI integrin and activation of uPA/uPAR, resulting in matrix degradation [134]. Another active participant in this process is the inhibitor of uPA—plasminogen activator inhibitor (PAI-1), which is released from the degraded matrix and internalized, further directing the balance toward invasive cell migration, i.e., angiogenesis (Figure 4). Inhibition or deficiency of uPAR suppressed VEGF-induced angiogenesis in tumor cells [135] or in mice [136], respectively. Moreover, uPAR stimulated angiogenesis through VEGFR2, which upon internalization activates other pro-angiogenic stimuli [136]. In confluent ECs, the expression of uPAR is down-regulated compared to sub-confluent proliferative cells, thus preventing VEGF-activated signaling and angiogenesis [137]. In addition, levels of PAI-1 are elevated in senescent and aged ECs, making it a useful marker for senescence [138]. Besides inhibiting uPAR, PAI-1 also induces p53 and p21, activity that is suppressed by SIRT1 overexpression in endothelial cells. SIRT1 is also able to induce eNOS activity, protecting ECs from endothelial dysfunction [138]. The pro-angiogenic properties of exosomes from ECs may also be attributed to EV-associated micro RNAs such as miR-214 [139]. More specifically, the latter prevents senescence through silencing ATM in recipient cells.

Depending on their source and the specific experimental conditions, EVs may also have anti-angiogenic properties. For example, NO production and angiogenesis are impaired by EC-derived EVs under oxidative stress, via Src kinase- and NOX-dependent mechanisms [66,140,141]. Moreover, in contrast to EVs from young cells, those derived from senescent cells exert mostly negative effects on EC functions and angiogenesis. More specifically, senescent osteoblasts secrete EVs that induce senescence and apoptosis and decrease proliferation of ECs through transfer of miR-139-5p [142]. Likewise, senescent HUVEC cells secrete exosomes that interfere with cell growth and downregulate expression of adherent junction proteins, resulting in impaired endothelial migration of young ECs and endothelial barrier dysfunction [143].

Interestingly, the effect of EVs on angiogenesis may be swayed in opposite directions depending on the dose. Namely, it was found that a low dose of EVs exhibited pro-angiogenic activity, which was suppressed below control levels upon increasing the concentration of the EC-derived EVs [144]—an effect dependent on uPA activity. The inhibitory effect of EC-derived EVs on endothelial cell tube formation was confirmed by another study in which even higher concentrations of EVs were used, and it was shown that the inhibition was dependent on NF-kB signaling and eNOS pathway suppression [145]. EVs are carriers of damaged genomic DNA molecules whose concentration increases in EVs upon induction of senescence [130] and under pathological conditions [146]. Functioning as intercellular vectors, EVs may transfer their DNA into the cytoplasm of recipient cells, leading to activation of the cGAS-STING signaling and consequently EC senescence, eNOS suppression and endothelial dysfunction [147]. Therefore, the hormetic effect of EC-derived EVs on EC tube formation, as well as the inhibitory effect of EVs from senescent ECs on angiogenesis, may possibly be due to EV DNA-induced cGAS-STING activation. Shedding more light on these processes and mechanisms would be a particularly interesting direction for further studies.

## 6. The Non-Productive Angiogenesis in Alzheimer’s Disease

Currently, there are two main hypotheses for the development of AD—the accumulation of amyloid plaques (Aβ) due to an error in the metabolism of the amyloid precursor protein (APP); and the hyperphosphorylation of Tau (or p-Tau), resulting in microtubule polymerization catastrophe and formation of fibrils [16]. APP is a transmembrane glycoprotein separated into an intracellular C-terminal, Aβ transmembrane and N-terminal extracellular domains. Its primary function is interneuronal communication, and once it performs it, APP is degraded by α- and γ-secretase to a soluble, non-amyloid form, or by β- and γ-secretase to insoluble Aβ_1–40_ and Aβ_1–42_ isoforms [16]. In animal models, elevated levels of Aβ_1–42_ and p-Tau were correlated with cerebrovascular dysfunction, chronic hypoperfusion and worsened AD symptoms [148,149]. One of the most affected brain areas in AD is the hippocampus, which is normally able to continue with adult neurogenesis. Thus, a decline in neurogenesis could be used as a marker for AD progression in animal models [150]. In fact, we demonstrated worsened long-term memory and anxiety in a rat model of icvAβ_1–42_ concomitant with pinealectomy (AD with melatonin deficiency). These behaviors are controlled by the hippocampus and corresponded with increased OS in the structure [108].

Pro-inflammatory cytokines, such as interleukin-1β, become abundant during AD and induce the expression of VEGF, yielding new blood vessels [18]. Although angiogenesis is initiated around Aβ plaques, the process is non-productive, leading to the disassembly of Aβ plaque-associated blood vessels and the phagocytic activity of microglia [151]. However, there is conflicting evidence relating the cause of AD and whether there is an increase or decrease in blood vessel density [151,152,153,154,155,156,157,158] (Table 2).

Joe Steinman, Hong-Shuo Sun and Zhong-Ping Feng provide a reasonable explanation for the discrepancies—“An overall measure of vessel density may indicate loss of vessels due to holes [note: from plaque deposits], without accounting for the increase in vessels surrounding holes” [159]. Although angiogenesis might not be beneficial for AD’s progression, it seems to alleviate some of the cognitive disabilities. For instance, one study showed that AD patients and AD mouse models accumulated Aβ in arterioles and experienced apoptosis of ECs [157]. When the same mouse model TgCRND8 was treated with VEGF, the growth factor was able to rescue vascular loss. And, most importantly, it significantly improved the behavior and memory of the subjects [157]. However, this observation could not be repeated in vitro on Matrigel^®^, where Aβ maintained low vascular density regardless of VEGF in tube formation assays, demonstrating the inability to always correlate in vivo and in vitro studies. A natural way to suppress Aβ accumulation is through melatonin. Besides its function as a radical scavenger, research shows that melatonin disrupts amyloid fibril formation [20] and exhibits anti-angiogenic properties [160,161]. Thus, by hindering Aβ plaque formation and reducing OS, melatonin deflects their role in non-productive angiogenesis and endothelial senescence. Taken together, these observations support the use of the hormone as an adjuvant therapy in AD.

### 6.1. How Does Aβ Stimulate Cerebral Angiogenesis?

As Aβ is produced by β- and γ-secretase from APP, there have been attempts to reduce Aβ production via enzyme inhibition. Unfortunately, this has led to dense and highly branched blood vessels. Cameron et al. [151] demonstrated that treatment of HUVECs and zebrafish with Aβ monomers and γ-secretase inhibitors increases the number of tip ECs and suggested an alternative mechanism through Delta-Notch signaling. Tip cell formation is supported by the interaction between EV-associated Delta like 4 (Dll4) and cell membrane-localized Notch, which restricts excessive sprouting angiogenesis, ensuring that only a limited number of cells will identify as tip cells and initiate new vessel formation [162]. Here, NAD^+^’s role is to improve VEGF sensitivity of tip cells and stimulate their proliferation, migration and ECM invasion, while hampering the transcriptional activity of Notch in the nucleus of stalk cells. Therefore, the latter cannot assume the functions of a tip cell and produce unnecessary sprouts [163]. Upon monomer binding to the Notch extracellular domain (NEXT), the same is cleaved to Notch intracellular domain (NICD) by γ-secretase and suppresses tip cell formation and hypervascularization. In the case of AD, Aβ competes with NEXT for proteolysis and counteracts the inhibition of neovascularization (Figure 5).

Excessive angiogenesis is observed with γ-secretase inhibitors [151], while immunization against Aβ protects against amyloid-Notch-induced vessel formation [164]. This non-productive angiogenesis exposes AD patients to a significant risk of cerebral amyloid angiopathy (CAA) [10,11,159]. CAA is usually caused by Aβ accumulation in the small arteries and capillaries in the brain, leading to chronic hypoxia, microaneurysms and dementia. As a confirmation, a study using magnetic resonance angiography showed disturbed blood flow in an 11-month-old APP23 transgenic mouse AD model compared to 20-month-old WT mice [11]. The researchers could not detect Aβ accumulation in larger arteries and assumed that the blood vessel disruption was due to soluble Aβ monomers. On the other hand, Aβ monomers were shown to exhibit pro-angiogenic effects, while Aβ oligomers triggered senescence in ECs through the p53/p21 pathway [155]. Aβ_1–42_ monomers or fibrils had no effect on p21; instead, they increased VEGFR1 and decreased VEGFR2 expression. The overexpression of VEGFR1 readily induced senescence in brain MVs. At the same time, siRNA against VEGFR1 prevented upregulation of p21 upon Aβ_1–42_ oligomer treatment. While some studies have proposed that VEGF could be sequestered in Aβ plaques, Alvarez-Vergara et al. [153] observed high expression of VEGF in astrocytes surrounding Aβ plaques and an association between VEGF expression and the protrusion of filopodia from endothelial cells in mouse models. The integrin αVβIII was also concentrated around the plaques, indicating vascular remodeling. A similar conclusion was drawn from single-nucleus transcriptome analysis of AD patients, which showed an induction of a subpopulation of ECs with increased expression of growth factors and their receptors [165]. In addition, the examined ECs acted as antigen-presenting cells by MHC-I, which typically indicates viral infections. It is reasonable for Aβ deposits to induce a pro-inflammatory response, but why MHC-I is involved in this process remains unclear.

Cells or animals can be genetically engineered with mutated APP or Tau proteins to generate in vitro and in vivo models of AD. Alternatively, AD can be induced by exogenous treatment with purified Aβ peptides. Even though, these peptides can be of various lengths and modifications, once they are solubilized, they are not stable, and improper handling can negatively impact the acquired results. Initially, the peptides turn into monomers, which associate together, forming oligomers, and finally to amyloid plaques, and all of these transitions are spontaneous in a water solution. Thus, the type of Aβ used for treatment should be validated and explicitly stated in a study, because the cells’ response varies depending on the applied Aβ form. Moreover, tissue samples acquired postmortem from animal models or AD patients must be stored and prepared adequately as soon as possible since many macromolecules can deteriorate and give false data in later examinations.

### 6.2. EVs Can Be Used as Biomarkers for Early AD Detection

EVs, and more specifically—exosomes, isolated from all sorts of AD model systems are shown to carry Aβ (or APP) along with exosome markers such as Flotillin-1 and Alix [166,167,168,169]. Moreover, some studies demonstrate a prion-like toxic activity of Aβ-carrying exosomes, where shortly after being endocytosed, Aβ starts to propagate and induce cell death [166,167,170]. Along these lines, many studies have attempted to use EVs as biomarkers for early detection and prognosis of neurodegeneration. In a recent study, Gallart-Palau et al. used brain EVs in the progressive course of AD and performed a proteome-wide analysis [171]. They found damaged mitochondria, APP and prion proteins (PrP) in EVs from the temporal lobe of AD patients due to impaired autophagy. What is interesting is that they rebutted the hypothesis that PrP and APP together exhibit neurotoxicity. Instead, their results show a co-upregulation of both PrP and APP at the preclinical stage of the disease, where PrP binds with APP and helps to sequester it in brain EVs (in agreement with another study [172]). Unfortunately, this protective mechanism deteriorates and is inefficient at the clinical stage of AD [171]. Nevertheless, these and many other studies demonstrate the role of EVs in disease progression and their potential as biomarkers [168,173,174,175]. It is unclear whether the size of EVs determines their mode of distribution between neuronal cells. Gabrielli and colleagues propose a mechanism according to which small EVs are endocytosed and spread their pathologic cargo trans-synaptically, whereas larger EVs move along the surface of axons, jump between connected neurons, and finally activate a signal and/or become internalized at synaptic sites [174]. In AD, hypoxia impedes the autophagy in neurons, causing the release of EVs carrying dysfunctional mitochondria and APP. While these hypoxic EVs can exhibit pro-neurodegenerative function, they can also supply mediators of hypoxia adaptation, angiogenesis and protein quality control [176].

### 6.3. The Role of Endothelial Progenitor Cells (EPCs) as a Biomarker and Potential Therapeutic Target in AD

There is increasing evidence that points to the alteration and dysfunction of the cerebral vasculature as an important factor in assessing the pathophysiology of AD, and this process may contribute to the onset of neurodegeneration, inflammation, Aβ accumulation and tau phosphorylation [177,178]. The so called two-hit vascular hypothesis proposed by Zlokovic and co-workers [179] suggested that damage in the cerebral vasculature (hit one) induces the accumulation of Aβ in the brain (hit two). In this respect, endothelial progenitor cells (EPCs) appear as a possible biomarker for early detection of AD as well as a therapeutic target given their role in maintaining the vasculature. EPCs, which are a rare population of cells originated from the bone marrow [180], circulate in the peripheral blood and have a capacity to repair or replace the damaged vessels. The most characteristic surface markers of EPCs are CD34, VEGF receptor 2 (VEGFR-2), and CD133. There are two different types of EPCs: (a) early-outgrowth EPCs (e-EPCs), circulatory angiogenic cells, or colony-forming unit endothelial cells (CFU-EC), which take part in the process of network formation and the repair of injured endothelial cells in a paracrine way by secreting different angiogenic factors; and (b) late-outgrowth EPCs (l-EPCs), endothelial outgrowth cells, or endothelial colony-forming cells, which improve angiogenesis by differentiating into mature endothelial cells [181]. In addition to their different functionality, both types of EPCs can be recognized by their appearance *in vitro*. Whereas e-EPCs appear after a few days in culture and form colonies with spindle-shaped cells around them, l-EPCs appear after 2–3 weeks in culture and present a cobblestone shape [182].

*EPCs as prognostic markers*—The number of EPCs and their ability to form CFU-EC colonies has been proposed as a possible marker of vascular function in AD [183]. In a clinical study of AD patients, Kong and co-workers [184] observed reduced numbers of circulating EPCs compared to healthy patients and that lower numbers of EPCs correlated with greater cognitive impairment. In addition, EPCs from moderate and severe AD showed functional alterations in culture (such as reduced adhesion and migration capacity) compared to mild AD and controls [183,185].

*EPCs as therapeutic target*—Additionally to the analysis of the number of EPCs in AD, the therapeutic potential of these cells has also been suggested in different animal models. For instance, when e-EPCs were injected intravenously into repeated scopolamine (SCO)-induced cognitive impaired rats, it resulted in improved learning and memory, attenuation of Aβ plaque deposition, as well as suppression of Aβ and p-tau levels. Similarly, when l-EPCs were injected intravenously into APP/PS1 transgenic mice, researchers observed an enhanced penetration of exogenous EPCs into the brain compared to controls. Subsequently, if l-EPCs were injected directly into the hippocampus of the same transgenic mouse model [186], they could lead to up-regulation of tight junction proteins (such as zonula occludens-1, occludin, and claudin-5) in the BBB, increasing microvessel density and promoting angiogenesis in the hippocampus and cortex. In addition, EPCs also showed an anti-apoptotic effect, promoting neuronal survival in the hippocampus. Other effects of EPC transplantation were the inspected reduction in the area and intensity of Aβ plaques in the hippocampus and cerebral cortex and significantly improved learning and memory in AD mice (APP/PS1). Recently, transfected EPCs that release antibodies against Aβ and reduce its aggregation have been generated [187]. Therefore, EPCs are postulated as a good therapeutic option for pathologies that present BBB alterations.

## 7. Therapeutic Approaches to Endothelial Senescence and Dysfunction

Understanding the underlying mechanism of aging and neurodegenerative diseases will one day provide us with the means to treat them. Along with DNA damage, OS, and insufficient or disturbed blood flow, behavioral and social cues guiding unhealthy lifestyle choices also accelerate the aging process. It is urban knowledge that chronic stress with high cortisol levels, high-calorie food, lack of exercise, etc. worsens life quality and expectancy. As presented in this review, regular exposure to hormetic stress can substantially improve vascular fitness, while properly controlled angiogenesis could delay both aging and neurodegenerative processes. Indeed, many approaches entail exercise and caloric restriction (CR) as therapies for vascular health instead of drug treatment.

### 7.1. Exercise Improves CBF, Vascular Function and Cognitive Performance

Angiogenesis in the brain microvasculature can improve tissue oxygenation, but if done improperly, it can provoke vascular leakage and neurodegeneration. A way to ensure positive angiogenesis is exercise, which stimulates eNOS by increasing the CBF [188] and potentially reduces OS by hypoxia-mediated inhibition of oxidative phosphorylation. In addition, aerobic exercise increases energy consumption (mimicking CR), while alleviating basal membrane dysfunction [189] and age-related behavior changes [190].

In an eight-week comparative study between old sedentary and exercised male rats, moderate exercise decreased the mean arterial blood pressure in favor of the trained group [188]. It also improved CBF, VEGF, eNOS expression, capillary density and astrocyte growth [188,191,192]. Furthermore, malondialdehyde (MDA—a marker for lipid peroxidation) levels were reduced in the exercised aged group [188]. Exercise also reduced the levels of fibrin and fibrinogen in old mice, improving the activity of neurovascular units (microvascular ECs, basement membrane, pericytes and astrocytes) [189]. Increased CBF, by regular treadmill running, prevented the loss of BDNF, which usually leads to learning and memory deficiencies [190]. Aerobic running on a treadmill or cycling induces EV secretion before reaching an anaerobic state [193]. In contrast, Brahmer et al. collected EV samples of athletes before, during and after cycling to exhaustion [194]. They observed a significant increase in CD63^+^ EVs post exercise (at the highest lactate levels), with some also carrying CD105 and CD146 (markers for ECs). Thus, exercise itself rather than the intensity influences EV release. The EV release is very likely to be Ca^2+^-dependent, and since muscle activation leads to Ca^2+^ flux, this could be a potential cause of EV accumulation [195]. Meanwhile, Ca^2+^ signaling is impaired in senescent ECs and impedes the contraction of vascular smooth muscle cells in mesenteric arteries of aged (24–26 month old) mice [196]. Taken together, these findings support speculation that the increase in plasma Ca^2+^ due to exercise could improve the vasomotor control of the arteries. Furthermore, exercise-induced moderate hypoxia causes metabolic conversion to anaerobic glycolysis, securing NAD^+^ availability when the TCA cycle and the ETC are subdued. The resulting buildup of lactate provokes the expression of VEGFR2 in ECs [197] and stimulates reparative angiogenesis in ischemic tissues [198]. Furthermore, lactate secreted by skeletal muscle can travel through the blood and bind to the lactate receptor HCAR1, enriched in cells lining the brain’s blood vessels, inducing VEGF expression and cerebral angiogenesis [191]. This was positively impacted by high-intensity interval training (HIIT) or lactate injections and led to increased capillary density in the brain of WT mice and not in HCAR1-KO. The authors linked this effect with the activation of ERK1/2 and Akt, which are upstream positive regulators of VEGF [191]. In general, physical activity improves physical and cognitive function by enhancing CBF and reducing OS, neuroinflammation and vascular dysfunction, and positively impacts AD’s symptoms.

### 7.2. Caloric Restriction Reduces OS and Vascular Aging

Already, Ciceron has suggested that moderate eating and exercise are key factors for longevity. Therefore, caloric restriction (CR) could be beneficial for people, as it activates autophagy and triggers the cells to recycle and renew themselves [199]. Under CR, high temperatures or excessive competition, *C. elegans* undergoes a dramatic metamorphosis into a dauer form. In this state, the worms close their mouth apparatus, switch their metabolism from the TCA cycle to gluconeogenesis and seize their development until food becomes available. The incredible thing is that dauers live at least twice as long compared to adult worms [200]. This is one of the reasons why *C. elegans* is the go-to system when studying senescence. However, the restricted activity of mitochondria ultimately leads to their deterioration [94]. In a recent study, mice meeting their caloric needs but consuming less protein and branched fatty acids had lower adiposity, higher metabolic rates and lifespans [201]. The authors attributed this to lower activation of mTORC1 by amino and fatty acids, rather than CR itself. With aging, mTORC1 is upregulated, which correlates with eNOS uncoupling and O_2_^•−^ generation, which are significantly reduced in senescent ECs treated with rapamycin (an mTOR inhibitor) [202] and in old mice under CR diet [203]. A detailed review by Christopher R. Martens and Douglas R. Seals describes other stress-induced cellular mechanisms inhibited in senile ECs—NO synthesis mediated by AMPK and SIRT1, autophagy (detailed review of autophagy factors promoting longevity [204]), and ECM stiffening through elastin proteolysis by MMP-9 and AGEs-induced inflammation of the arterial wall that can be ameliorated by CR [205]. Furthermore, the activity of SIRTs as histone deacetylases, hence, the epigenetic regulation of senescence and aging, is promoted by CR [178]. Although there is substantial evidence that CR can reduce and delay the deteriorating effects of aging and maintain our longevity, more controlled research is necessary to establish good CR protocols accounting for personal needs.

### 7.3. Role of Resveratrol in the Vascular Biology and Senescence Process

In general antioxidants such as reduced glutathione (GSH) and melatonin inhibit cell senescence by reducing reactive oxygen species (ROS) generation [206]. Resveratrol (3,5,40-trihydroxystilbene) (R), which is a non-flavonoid polyphenolic compound and derivative of stilbene, exhibits its pleotropic function also by decreasing ROS production and improving the antioxidant levels [207]. As mentioned above, EPCs are critical circulating components of the endothelium and are identified as key factors in endothelial repair. In this respect, resveratrol treatment can reverse EPC dysfunction by decreasing oxidative stress and increasing proliferation and capillary-like structure formation, and, by increasing the angiogenic factors like (NO), can reverse stress-induced senescence [208].

## 8. Conclusions

The study of angiogenesis in the context of endothelial senescence, aging and Alzheimer’s disease has revealed their intricate complexity and heterogeneity. Even though, senescent cells can trigger inflammation, they can also support tissue renewal in the adult organism. Their dual role depends on the time, place and degree of their accumulation. The induction and outcome of endothelial senescence can vary across different cell types, but it underlies vascular dysfunction and subsequent non-productive angiogenesis and vascular leakage. Short hormetic stress employed on blood vessels by hypoxia, metabolism switch or high shear stress can reduce OS and improve EC responsiveness to angiogenic stimuli and cognitive function. Identifying universal senescence markers remains a challenge, and careful selection and consideration of their limitations are crucial for accurate research conclusions. Furthermore, senile ECs secrete SASP factors that can accelerate aging and neurodegeneration through induced inflammation. As a key SASP component, EVs can be used as biomarkers for the early detection of AD, and the development of standardized repositories for SASP markers could enhance their application and reproducibility. Furthermore, studying neurodegenerative diseases and angiogenesis, researchers must choose suitable models and consider factors such as the type of Aβ peptide and endothelial cell line since the responses can vary significantly. Innovative in vitro and in vivo models could provide more physiologically relevant insights. Addressing these considerations will contribute to advancing our understanding of endothelial senescence and related processes.

## Figures and Tables

**Figure 1 ijms-24-11344-f001:**
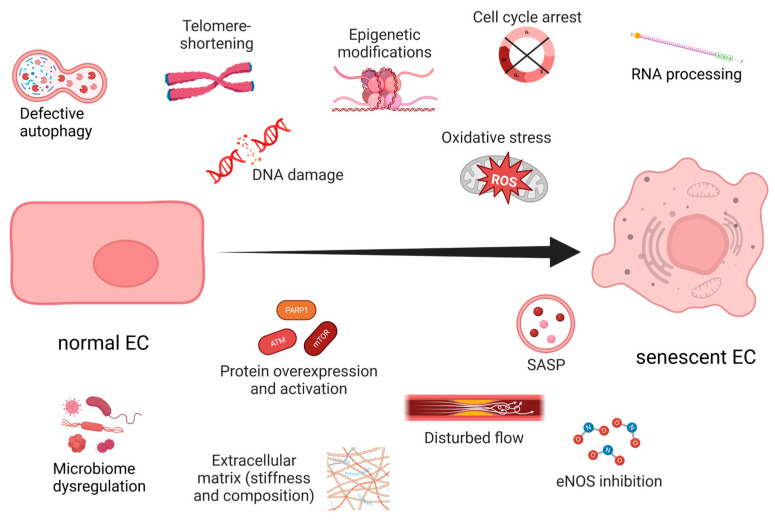
Hallmarks of endothelial senescence. Created with https://www.BioRender.com (accessed on 7 July 2023).

**Figure 2 ijms-24-11344-f002:**
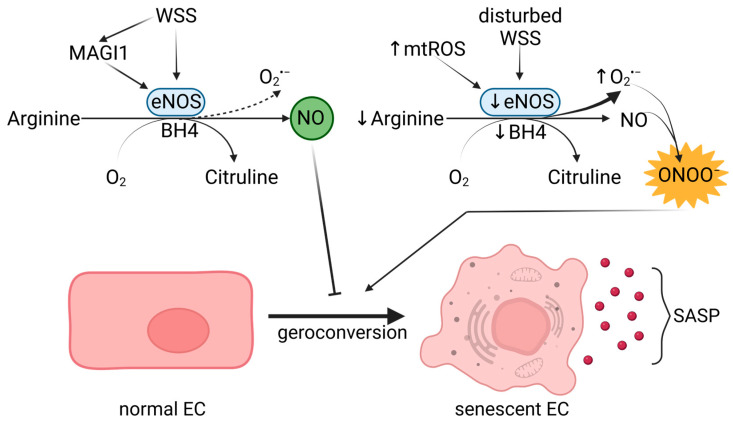
The role of eNOS in endothelial senescence. Wall shear stress (WSS) and MAGUK with inverted domain structure-1 (MAGI1) activate eNOS, which produces NO from Arginine. The accumulated NO inhibits the geroconversion of ECs. When the levels of tetrahydrobiopterin (BH4) or Arginine are low, eNOS’s function is altered, resulting in its uncoupling and the increased production of superoxide anion (O_2_^•−^). The latter reacts with NO and yields ONOO^−^ (peroxynitrite), which causes senescence in ECs. eNOS is also negatively impacted by disturbed WSS and mitochondrial ROS. Senescent ECs generate SASP, which can further exacerbate endothelial dysfunction. Created with https://www.BioRender.com (accessed on 11 July 2023).

**Figure 3 ijms-24-11344-f003:**
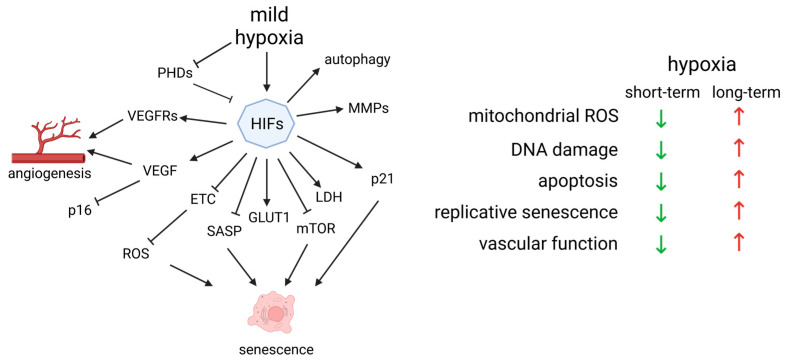
Summary of the effects of short-term (mild) hypoxia on EC signaling. Short-term hypoxia reduces mitochondrial ROS, DNA damage, senescence and apoptosis, protecting the endothelium. During mild hypoxia, hypoxia-inducible factors (HIFs) are activated, unlike in normal oxygen levels when they are targeted by prolyl hydroxylases (PHDs) for degradation. HIFs stimulate VEGF and its receptors (VEGFRs), and metalloproteinases (MMPs), which stimulate the reorganization of the extracellular matrix and follow-up angiogenesis. The cell’s renewal through autophagy is also activated. HIFs induce anaerobic glycolysis by glucose transporter 1 (GLUT1) and lactate dehydrogenase (LDH), circumventing the ETC and further reducing OS. HIFs can cause overexpression of p21, which leads to senescence, but they inhibit multiple other pro-senescent factors such as p16, senescence-associated secretory phenotype (SASP), and mammalian target of rapamycin (mTOR). Red arrows indicate an increase in the corresponding processes, which have a negative senile effect on the cells. Green arrows indicate a decrease in said processes, which protect the cells from senescence. Created with https://www.BioRender.com (accessed on 30 May 2023).

**Figure 4 ijms-24-11344-f004:**
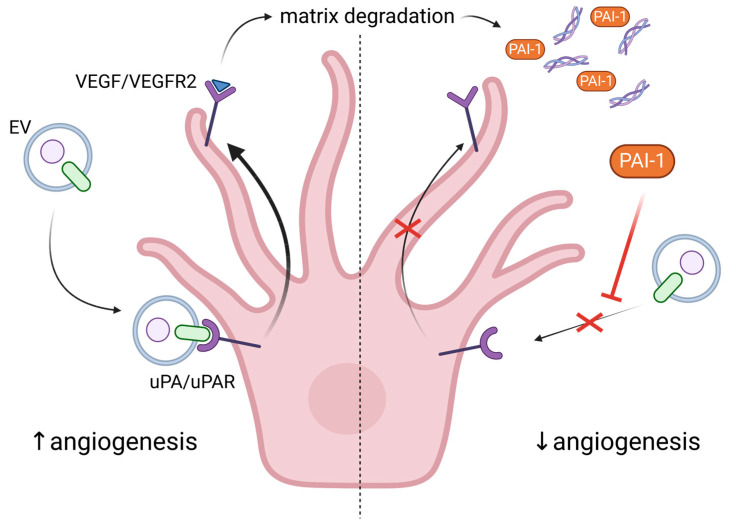
Regulation of angiogenesis by urokinase plasminogen activator and its receptor. uPA/uPAR are carried by extracellular vesicles (EVs) to ECs. Upon receptor binding, VEGF-mediated matrix degradation is stimulated via VEGFR2. The processed matrix releases the plasminogen activator inhibitor-1 (PAI-1), which inhibits uPA/uPAR recognition and subsequent VEGFR2 activation. This feedback loop prevents excessive angiogenesis. The red X depicts the obstruction of uPA/uPAR recognition under the influence of PAI-1 and the inability of uPA/uPAR to activate VEGFR2. Created with https://www.BioRender.com (accessed on 30 May 2023).

**Figure 5 ijms-24-11344-f005:**
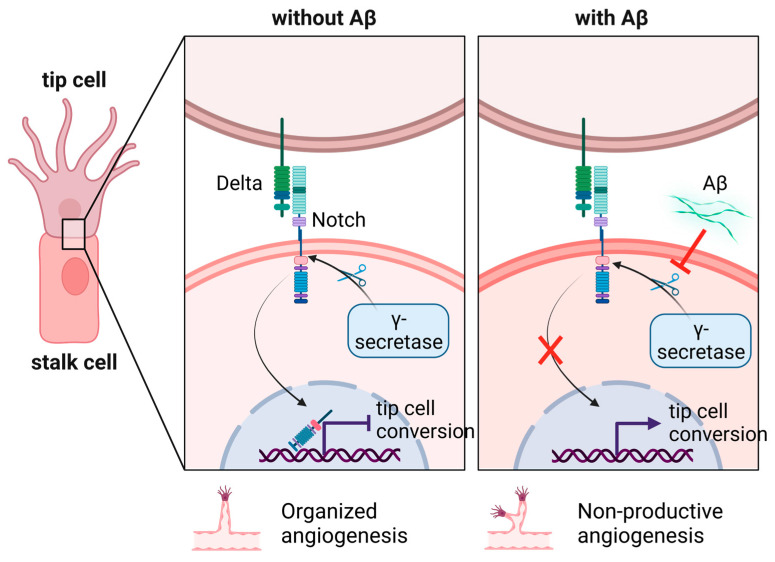
Regulation of tip cell formation through Delta-Notch signaling. Under normal conditions, Delta-Notch signaling serves in intercellular communication. Upon their binding, Notch is cleaved by γ-secretase into an extracellular domain (NEXT) and an intracellular domain (NICD). The latter acts as a transcription factor and inhibits genes involved in tip cell conversion in stalk cells. The amyloid protein Aβ serves as a competitive inhibitor of Notch and distracts γ-secretase. As a result, NICD cannot inhibit tip cell conversion of stalk cells and triggers non-productive angiogenesis. Created with https://www.BioRender.com (accessed on 30 May 2023).

**Table 1 ijms-24-11344-t001:** Effects of the type of wall shear stress (WSS) on EC function.

Type of WSS	Cell Line	Cell Response	Reference
Sudden, temporal, 10 dyne/cm^2^	HUVECs	↑ proliferation	[74]
Steady, uniform	HUVECs	no effect	[74]
Linear, physiological, 12 dyne/cm^2^ *	BAECs	↓ proliferation	[75]
Gradient, <68 dyne/cm^2^	HMVECs	migrate against flow; orient perpendicularly at highest WSS	[80]
Linear, high, 284 dyne/cm^2^ Gradient, positive, 150–170 dyne/cm^2^	BAECs **	↓ alignment; ↑ proliferation; ↑ apoptosis	[81]
Linear, low, 30 dyne/cm^2^ Gradient, negative, 170–150 dyne/cm^2^	BAECs	↑ alignment; ↓ proliferation; ↓ apoptosis	[81]

* Physiological WSS 10–20 dyne/cm^2^; 10 dyne/cm^2^ = 1 N/m^2^ = 1 Pa. ** BAECs—bovine aortic endothelial cells. ↑—indicates an increase in the process; ↓—indicates a decrease in the process.

**Table 2 ijms-24-11344-t002:** The role of Aβ in cerebral blood vessels.

AD Model	Blood Vessels	Protein Expression	References
Aβ monomers in HUVEC and zebrafish	↑ capillary density	-	[151]
Tau overexpressing mice; 15 months old	↑ capillary density; ↑ angiogenesis; ↑ BBB permeability; ↓ CBF	↑ VEGF; ↑ uPAR; ↑ MMP-9; ↑ PAI-1	[152]
AD patients	−	↑ VEGF; ↑ TGF-β	[153]
HMVECs + Aβ monomers	↓ angiogenesis	↑ VEGFR1- ↑ senescence	[154]
APP-PSEN1/+ mice	↑ non-productive angiogenesis; ↓ capillary density around plaques	↑ VEGF	[155]
Tg2576 mice	↓ capillary density around plaques	↓ GLUT1	[156]
AD patients; APP695 mice	↓ capillary density	VEGF supplementation improved cognitive function	[157]
3xTG-AD mice	↑ capillary density; ↓ junction density	−	[158]

↑—indicates an increase in the process; ↓—indicates a decrease in the process.

## Data Availability

Not applicable.

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
