# Peer review of "Endothelial Senescence and Its Impact on Angiogenesis in Alzheimer’s Disease"

_ijms, 2023, doi:10.3390/ijms241411344_

Round 1

Reviewer 1 Report

In this article by Georgieva at al., the authors present an extensive overview of recent research on endothelial senescence and its functional consequences. They thoroughly discuss the complexity of factors that affect senescence of endothelial cells in context of its association with processes of angiogenesis and neurodegeneration. The review is an interesting and much-needed contribution to the field.

The specific points that need to be addressed prior to publication:

1.     In chapter 2.1, the authors should rather talk about different factors affecting cell senescence than about “theories of aging”.  There are no unified or clearly defined theories of aging, thus described in the chapter 2.1 “three main theories of aging” are concepts that do not exclude each other but comprise of the connections among factors contributing to the senescent phenotype.

In this case, the authors may consider the citation of the below article:

Schmauck-Medina T, Molière A, Lautrup S, Zhang J, Chlopicki S, Madsen HB, Cao S, Soendenbroe C, Mansell E, Vestergaard MB, Li Z, Shiloh Y, Opresko PL, Egly JM, Kirkwood T, Verdin E, Bohr VA, Cox LS, Stevnsner T, Rasmussen LJ, Fang EF. New hallmarks of ageing: a 2022 Copenhagen ageing meeting summary. Aging (Albany NY). 2022 Aug 29;14(16):6829-6839. doi: 10.18632/aging.204248.

2.     Similarly, the authors should be cautious using the term “theories” in the title “2.1. Theories of aging”.  They should rather use the title “2.1. Hallmarks of aging”, as they stated in the description of corresponding Figure 1.

3.     Given that p53/p21 and p16/pRB signaling pathways are known to act mainly independently in cellular senescence, p53 can not be described as an upstream regulator of p16. Thus, the below sentences need to be adequately corrected:

“In the case of chronic stress, p53 triggers senescence through either p21 or retinoblastoma (Rb)/p16 pathways [24,57].”

“P53 is at the center of senescence signaling as it can trigger it through p16 or p21”.

Minor points:

1.     I believe that the word “Title 4” in the heading of Table 2 should be exchanged for the word “Reference”.

2.     Figure 2 is not mentioned in the text.

Quality of English language is good.

Reviewer 2 Report

The current review tackles an interesting topic, namely, the implications of endothelial senescence for Alzheimer's diseases. The review is nicely written but wordy to some extend also illustrated by more than 200 references which seems excessive. It would benefit from a shortening and a focus on the actual topic.

Figure 1 seems problematic as it is too close to the famous "Hallmarks of cancer" reviews in Cell by Doug Hanahan and Bob Weinberg. This renders it close to plagiarism.

Minor points:

Line 50: ECs normal function should be ECs physiological function. Normal is an inappropriate judgement.

Line 205: cell’s behavior is wrong as behavior is a feature of complex organisms not cells

Lines 402-405: Though interesting the paragraph seems unscientific to some extend

Ok

Reviewer 3 Report

-The abstract must be completed with a better introduction to your subject: neurodegenerative diseases and Alzheimer's disease in particular. Also, please specify, that this is a narrative review.

-In introduction section, it would be interesting to include a paragraph on neurodegenerative diseases and Alzheimer's in particular.

-In section theories of aging, it would be helpful to differentiate "aging" from "senescence". It would also be good to better describe the stress-induced premature senescence (SIPS) often associated with oxidative stress and regulated by sirtuins. And notably SIPS can be reversible by calorie restriction as you mentioned in “therapeutic approaches against endothelial senescence and dysfunction”.

-Endothelial progenitor cells (EPCs) have an essential role in the endothelial repair processes, and in vascular alerations in alzheimer’s disease (Front. Aging.Neurosci, 26 january 2022. https://doi.org/10.3389/fnagi.2021.811210. Please discuss this point. 

-In therapeutic approaches against endothelial senescence and dysfunction section, please include a section concerning pharmaceutical approach and notably the role of resveratrol in the vascular biology and senescence process (Int. J. Mol. Sci. 2023, 24(11), 9747 https://doi.org/10.3390/ijms24119747).

-There are a lot of references, would it be possible to reduce their number and choose the most relevant and recent ones?

Round 2

Reviewer 3 Report

the answers provided by the authors are satisfactory and the additions incorporated to the manuscript important. Therefore, in my opinion, this manuscript can be accepted for publication.